# Parental experiences of the impacts of Covid-19 on the care of young children; qualitative interview findings from the Nairobi Early Childcare in Slums (NECS) project

Robert C. Hughes[1]*, Ruth Muendo[2], Sunil S. Bhopal[1,3], Silas Onyango[2], Elizabeth Kimani-Murage[2], Betty R. Kirkwood[1], Zelee Hill[4‡], Patricia Kitsao-Wekulo[2‡]

1 Department of Population Health, London School of Hygiene & Tropical Medicine, London, United Kingdom, 2 Maternal and Child Wellbeing Unit, African Population and Health Research Center, Nairobi, Kenya, 3 Population Health Sciences Institute, Newcastle University, Newcastle upon Tyne, United Kingdom, 4 Institute for Global Health, University College London, London, United Kingdom

‡ ZH and PKW are joint last authors.
* Robert.Hughes@LSHTM.ac.uk

## Abstract

### Introduction

The Covid-19 pandemic, and societal attempts to control it, have touched almost every aspect of people's lives around the world, albeit in unequal ways. In particular, there is considerable concern about the way that stringent 'lockdowns', as implemented in Kenya and many other countries, affected young children, especially those living in informal settlements. However, to date, there has been little research attempting to unpack and understand how the pandemic has impacted on the care of young children.

### Methods

In-depth telephone interviews were conducted with 21 parents/carers of children aged under five years living in three Nairobi slums between May and September 2021 exploring the ways in which Covid-19, and policies to control the pandemic, impacted on their household and the care of their child/children.

### Results

The impacts of Covid-19 control measures on the care of children have been widely felt, deep and multiple. The impact of economic hardship has been significant, reportedly undermining food security and access to services including healthcare and childcare. Respondents reported an associated increase in domestic and community violence. Many people relied on help from others; this was most commonly reported to be in the form of variable levels of flexibility from landlords and help from other community members. No direct harms from Covid-19 disease were reported by respondents.

**Data Availability Statement:** We are happy to make available to all qualified researchers on request the underlying data drawn on for this research. Given the challenges of genuine anonymisation of qualitative data, we have not posted the data to an open repository, but we are happy to facilitate sharing requests via email to researchdatamanagement@lshtm.ac.uk.

**Funding:** This work was supported by the British Academy (Grant number ECE190134) and Echidna Giving who supported RCH through a linked Clinical Research Fellowship. SB is supported by a NHIR clinical lecturership at Newcastle University. BK, ZH, PK-W, SO and RM received partial salary support from the British Academy grant, and RCH received partial salary support from Echidna Giving. The funders had no role in study design, data collection and analysis, decision to publish, or preparation of the manuscript.

**Competing interests:** The authors have declared that no competing interests exist.

## Conclusion

The impacts of Covid-19 control measures on the care of young children in informal settlements have been indirect but dramatic. Given the breadth and depth of these reported impacts, and the particular vulnerability of young children, deeper consideration ought to inform decisions about approaches to implementation of stringent disease control measures in future. In addition, these findings imply a need for both short- and long-term policy responses to ameliorate the impacts described.

## Introduction

The Covid-19 pandemic has touched the lives of almost everyone on the planet, but in very different ways. In Kenya, there was an early and stringent response to the first cases of community transmission, including one of the most harshly enforced 'lockdowns' in the world [1]. Efforts to control the pandemic in Nairobi were particularly felt in the informal settlements where 60% of the city's population lives [2]. Enforcement of lockdowns was strict, with reports of violence and heavy-handed crackdowns from police especially in informal settlements [3].

Early childhood is a critical window of opportunity; adversity in this period is a central social determinant of health and wellbeing, affecting later life learning, earning and happiness [4], The 2018 joint WHO/UNICEF/World Bank "Nurturing Care" Framework (illustrated in Fig 1), describes five–intersecting–domains or components which can support healthy early childhood development: good health; adequate nutrition; responsive caregiving; opportunities for early learning; and safety and security [5]. While early childhood is described differently by different organisations, for the purposes of this research we have focused on the period until a child's fifth birthday.

The direct effects of Covid-19 on children in Kenya, and the wider region, are poorly documented due to limitations in testing, reporting and healthcare systems [6]. That said, overall, the direct effects–despite the prevalence of immunodeficiency due to malnutrition and HIV–appear to be limited, with a recent UN Inter-agency Group for Child Mortality Estimation concluding that the direct effects of Covid on child mortality 'remain very mild' [7]

Early in the pandemic, concerns were raised about how the control measures would be likely to impact on young children, anticipating that "vulnerable children will bear the biggest brunt of the direct and indirect impacts of the pandemic". Shumba and colleagues (2020) noted that in addition to direct health impacts from Covid-19, young children are also at risk from impacts on health, nutrition, social and child protection systems alongside economic disruption [8]. However, little research has been published to date attempting to explore the lived experiences of these impacts as they have emerged, especially in low- and middle-income countries. In particular, we are unaware of any other research which has sought to gain an in-depth understanding of parents'/carers' experiences of the impacts of Covid-19 on the care of young children living in urban slums in sub-Saharan Africa.

We aimed to contribute to addressing this research gap through conducting in-depth telephone interviews with parents/carers from across three slums in Nairobi, Kenya, to gain an understanding of their experiences of caring for a child in this context at a time of Covid-19. The results presented here are part of the larger Nairobi Early Childcare in Slums (NECS) study which through mixed-methods sought to understand the use, provision and quality of paid childcare in an informal settlement in Kenya [9].

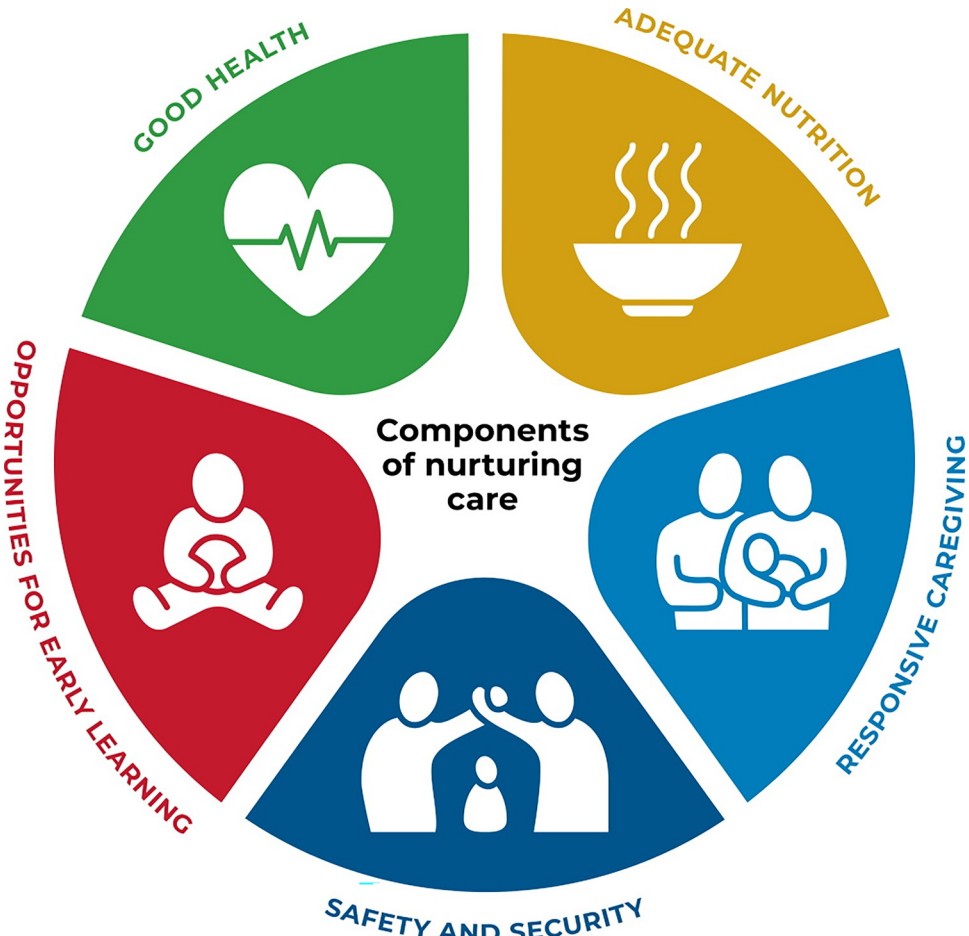

**Fig 1. The five domains of nurturing care, republished from [5] under a CC BY license, with permission from WHO, original copyright 2018.**

## Methods

### Study design

Qualitative in-depth interviews, conducted remotely by telephone.

### Timing, setting and participant characteristics

In-depth telephone interviews were conducted between 11[th] May and 17[th] September 2021, with parents/carers of children aged under five years who were living in one of three slums in Nairobi (Kibera, Kawangware and Mukuru-Viwandani). At this time, Kenya experienced a fourth wave of Covid-19, with the 7-day average number of reported cases ranging between 263 and 1974 [10]. At the time of the interviews, Covid-19 control measures were ongoing, albeit much less stringent than early in the pandemic. These consisted of night curfews (from 8pm and then later from 10pm), mask mandates in public areas, limits on public gatherings and advice to work from home [11]. In the preceding months much more stringent controls were in place, including widespread closures of workplaces and schools, 'stay at home' orders, movement restrictions and re-organisation of public health services towards Covid-19 treatment; a detailed chronology of these evolving controls is described by Herman-Rolloff et al. [12].

The three slums were selected primarily because collectively they are typical of the larger and longer established slums across Nairobi in which the majority of the population live. Practical considerations were relevant too; this sample was drawn from an existing database of telephone numbers for low-income households who had agreed to being invited to take part in research that our data collection partner, BUSARA [13], had previously collated.

These three slums are characterised by widespread poverty, poor water and sanitation provision, inadequate shelter, insufficient infrastructure, high levels of insecurity and high rates of informal employment [14]. All three slums are well established and have existed for decades. The ethnicity across all is mixed (including significant populations of Kikuyu, Luo, Luhya, and Kalenjin) as is the mix of new arrivals (including rural-urban migrants and international migrants) and long-standing residents. Each slum is loosely divided into villages, which tend to be dominated by one ethnic group, and boundaries are frequently blurred. Data on employment and education enrolment in the slums are limited, but a recent study found that approximately 50% of the population in Viwandani had completed secondary or more schooling and around a third of females and 8% of males were unemployed, noting that most work in the informal sector [15].

## Data collection

RM, an experienced interviewer with Masters level training in Development Studies, conducted telephone interviews in Kiswahili using a semi-structured topic guide developed by all authors (S1 Appendix). The content of the topic guide included both questions about Covid-19 impacts alongside a broader set of themes about childcare in the slums (manuscript in preparation). This topic guide was informed by a rapid review of the emerging literature on Covid-19 and nurturing care, including the work conducted by Shumba and colleagues [8] which considered how the domains of the Nurturing Care Framework (health, nutrition, responsive caregiving, early learning, and security and safety) would be affected by Covid-19 and/or pandemic control measures. The topic guide was initially drafted in English, then translated into Kiswahili (by RM). It was then back-translated in a meeting between RM, PK-W, RCH and SO to discuss any differences in meanings, and the best way to phrase questions. Interview topic guides, and the emphasis on different areas, were iterated over the course of data collection, building on experiences, perceptions and ideas that emerged. This was based on the discussions at weekly team reflexivity meetings, where emerging themes were discussed in order to identify areas where the team felt deeper exploration might be informative, and/or where alternative phrasing of questions might work better. For the most part, this related to how and when prompts were used to gain deeper insights into emerging themes, rather than adaptations to the topic guide.

Selection of participants was as follows. First, a list of respondents who had completed up to five rounds of the NECS Covid impacts tracker sub-study, (a bi-monthly structured telephone survey tracking the impact of Covid-19 on the care of children in slums [16]) was randomly ordered. Next, RM worked through this list, selecting participants purposively, including a mixture of both male and female parents/carers of a variety of ages of children, and both users and non-users of paid childcare. When participants with specific characteristics were sufficiently represented in the sample, other potential participants on the list were skipped until a participant with a desired characteristic was reached.

Timing of telephone interviews was pre-arranged through a recruitment call, during which respondents indicated whether they wanted to take part in the research; amongst those reached, none declined to participate after the study was explained to them. Participants were asked to find a quiet place to take the interview call. Calls started with RM introducing herself

and reading participant information and consent scripts. Where necessary, this information was re-phrased to improve clarity and ensure participant understanding. Any emerging questions were answered.

Interviews were digitally audio recorded. They were then simultaneously transcribed and translated verbatim from Kiswahili into English by a professional translator. Batches of 1–3 translated transcripts were reviewed and, where needed, corrected by RM in advance of analysis. RCH, RM, PK-W, SO and ZH met approximately weekly during fieldwork to review transcripts and field notes and identify and discuss emerging themes,. In addition, these meetings were used to discuss the emergence of saturation, when it was felt that further interviews would be unlikely to lead to additional insights. No repeat interviews were carried out.

**Public involvement.** Community engagement meetings were held in Mukuru in advance of the broader NECS Study in February 2020, introducing the study, and explaining the choice of research methods and the rationale for the research. Because this was before the Covid-19 pandemic, these did not specifically discuss the issue of pandemic impacts/controls. During preparation of this manuscript emerging findings were shared in a community meeting in Nairobi in March 2022, with a focus on how the findings can inform the work of both the county government and also community-based organisations.

## Ethical considerations

At the start of interviews, an information script including the rationale for the study, the voluntariness of participation, and information on data handling/sharing (see S1 Appendix) was read out and participants were asked to confirm that they agreed (1) to take part, (2) for the conversation to be recorded, translated and transcribed, and (3) for these data and results to be shared and used with researchers and others both in and outside of Kenya. This verbal consent process was audio-recorded. The LSHTM Research Ethics Committee (LSHTM Ref: 22692) and Amref Health Africa's Ethics and Scientific Review Committee (ESRC) in Kenya (Ref: P777/2020) reviewed and approved the study protocol. The National Commission for Science, Technology and Innovation (NACOSTI) provided research clearance. Participants were provided with a modest (equivalent to USD 3) talk-time credit after completion of the interview, as a contribution towards their expenses, for example battery charging for their phone.

## Data analysis

Data analysis was concurrent with data collection through regular weekly team discussions and a combination of iterative and deductive coding. Transcripts were read several times to build familiarity with the data and were then coded by RCH using NVivo 12 [17]. This started inductively, based on the participants' responses to the initial open questions about how the pandemic had affected them and the care of their children. Further sub-themes were then identified. These were then considered in relation to the domains of the Nurturing Care Framework [5] and/or as cross-cutting, and key themes were identified. The relationships between key themes, deductive Nurturing Care Framework domains and inductive codes is illustrated in S1 Fig.

Throughout, the focus was on understanding the underlying meaning behind statements and identifying widely held or contradictory responses/themes. Sub-themes and draft coding schedules were shared and discussed at regular intervals amongst the authors, and reflective notes were kept throughout the process. Through these discussions, the key themes presented in the results were identified.

**Researcher reflexivity.** The epistemological position of the researchers was discussed before and during analysis; with the team adopting a pragmatic position [18], seeking to focus

on the utility of knowledge to inform policy, programmes and interventions. RM, SO and PK-W are mixed methods early childhood development researchers living and working in Kenya. RCH, ZH, SB and BK are UK-based child health and development researchers. RCH has worked as a health adviser at several international donor organisations. SB is a practising community child health physician. EK is a mixed methods public health/nutrition Kenyan researcher with extensive experience in research urban poor settings in Kenya. All authors, being based in albeit inter-disciplinary, health research organisations bring biomedical experience/perspectives to this research, although they all work on social determinants of health. Our frequent meetings to discuss fieldwork and themes allowed us to reflect on the data as a team which, given our varied backgrounds and experiences, enabled us to reflect on how our backgrounds informed our interpretations. These included reflective discussions with RM about how her positionality, especially conducting interviews by telephone, how this may impact respondents and how this could be mitigated, for example through considering the timing/scheduling of calls, investing appropriate amounts of time in introductions and building rapport.

## Results

A total of 21 interviews were conducted. These took between 14 and 39 minutes, including the broader ranging discussion about childcare in slums but excluding the informed consent process. The mean duration was 22 minutes. All of the participants approached agreed to take part in the study.

The characteristics of the sample are described in detail in S1 Table. In summary it comprised 11 mothers, 8 fathers and 2 grandparents with similar number of users (n = 11) and non-users (n = 10) of paid childcare. Around half of the participants had children aged 12–23 months (n = 10).

Analysis identified three key themes. Firstly, indirect impacts of Covid-19 controls were more significant than reported direct effects of the virus. Secondly, these impacts were broad, and affected all domains of nurturing care. Finally, help, where it was available generally came from within the community rather than from the government.

### Indirect impacts of Covid-19 controls were more significant than direct effects of Covid-19

The first major theme identified was that the indirect impacts of Covid-19 control measures, in particular economic hardship, were more significant than the reported direct effects of Covid-19. The impacts of efforts to control the Covid-19 pandemic on the care of children in slums were described as significant and multi-faceted by all respondents. There was a universal sense that the pandemic had affected people and their daily lives deeply. However, all of the effects were indirect; although we did not directly ask about Covid-19 infections, when asked how the pandemic had affected their lives, none of the interviewees described knowingly suffering from Covid-19 infection themselves, or their children becoming unwell with the disease.

Economic effects were described by almost all respondents, with the loss of jobs and of informal income-generating opportunities affecting those working in a variety of roles and sectors, including domestic work, factory work, market trading and informal 'piece work' or daily labouring. All of these became even less reliable sources of income:

> Money has reduced. There is no money [but] needs are still many. . . .those things, even paying the house, has become a problem. . .. Because there is no way you will get to pay you

have to struggle. . . and sometimes you find you don't get. Surviving means doing any work that you will get–IDI15, father of a 20-month old user of paid childcare.

The economic impacts were described as cross-cutting, and affecting the whole community, at times leading to evictions, loss of household assets or changes in income-generating activities: People are indoors so there are no jobs. . . . We have hope you know. [But it is a] hard life. When the economy is down everyone is affected, we take home what we get and the costs rises. . . Things are not good, sometimes you will find some friends lost their jobs and sometimes they want a handout and maybe you don't have. And sometimes you are late on paying rent. [When you are unable to pay rent] They take someone's things or they close the house–IDI2, grandmother of a 18-month and 3-year-old who used to use paid childcare until they lost their jobs during the Covid epidemic

As a result of these economic impacts, some people described being forced to move, either to their ancestral village if they had the means to get there, or to a cheaper, often smaller or less well-located, house within Nairobi's slums:

Life was very expensive, now it became very expensive to pay for rent. . . So, we had to find a cheaper life that we can sustain–IDI17, male user of paid childcare for his 18-month and 4-year-old

These indirect effects of the pandemic were influenced by gender too. Male respondents frequently reported being especially responsible for earning money for the household, and females, including girls not attending closed schools, were more commonly responsible for childcare. Respondents described how the upheaval caused by the pandemic exposed some of these pre-existing expectations, and in some cases disrupted them, for example the crisis necessitating both parents to earn money for the household.

I was able to provide but since I lost my job life became very expensive. I stayed in the house for long thinking of what to do. You have lost your job and you don't have any other way, you don't have anywhere to go. You don't know how it is out there. Paying the house has become so hard you sometimes could stay for two or three months without paying rent. You have been given a notice. . . you don't know where you are moving to. You see those are the challenges. You know that time [when I was employed] I didn't have such challenges; I knew every month I have a salary and I also knew my family was catered for. I knew I am providing you see? So, when I lost my job, it became trial and error -IDI17

## Impacts on young children span all domains of nurturing care

Secondly, the impacts of Covid-19 controls on young children spanned all domains of nurturing care and were described in a variety of ways. Cutting across the domains, some parents/carers described the impacts of their own fear and a combination of confusion and some community denial about the epidemic, especially early on. For example, some reported how this led to them 'shielding' their children at home because of a combination of a strict interpretation of the restrictions and their own fear of the virus. This limited travel both within and beyond the city, reducing family and peer interaction, and leading to delaying or avoiding seeking of healthcare or shopping for food.

People were fearing even to go to someone's house or even greeting them. . . It happened that everyone was staying in the house and they don't want to go out, you only run to the shop or fetch water and go back in the house–IDI17

Some parents/carers described how they were especially worried about their children's risks from Covid-19 during the pandemic. This was either because of their desire to socialise with peers or because they were unable to use personal protective equipment that was thought to be effective like face masks:

[I am afraid to travel] because she can't put on a mask. . . So when I put on a mask to protect myself what of her? She will not allow me to cover her. She wants to look at everything–IDI12, mother of a 1-year-old, who stays at home with her

**Health.**   Although no respondents reported Covid-19 making members of their household or extended household unwell, disruption to health services was a concern, for example with some child health clinics being converted into Covid-19 treatment or isolation centres leading to them not taking children, especially for health promotion and prevention interventions:

But for now, [name of clinic] is for Corona patients. . . That is where we were taking a child for clinic–IDI12

Restrictions were described as placing considerable strain on all members of the household, including contributing to stress amongst parents/carers, with knock-on effects on the care of young children; discussed further below.

**Nutrition.**   In addition, significant knock-on effects of economic stress on food security were reported. Several study participants reported cutting down on meals to once a day, reducing the variety of food or relying on help from neighbours. The effects of this food insecurity were described as especially significant for children, accompanied by a sense of helplessness or lack of options. The challenge of managing on a day-to-day basis was clearly described:

. . .We suffered, we stayed without. Sometimes we would take strong tea without sugar and the child will not drink . . . He would cry for the whole day but what can we do? You wake up in the morning you don't have money and you find someone who gives you twenty shillings and you go and buy vegetables for ten shillings, a five-shilling tomato and an onion of five shillings and you add a lot of soup and you eat it–IDI5, father and user of paid childcare for 6-month old

He would eat yoghurt and chips and all these things stopped . . . Now he just eats what has been found, strong tea. . . And you can see the sadness in his eyes . . . when he asks for something and you are unable to provide–IDI7—mother of 4-year old childcare user

**Responsive caregiving and early learning.**   Lockdown was described as mostly undermining both peer-to-peer and parent-child interactions, especially as restrictions became protracted. This included children being unable to play with their peers and becoming bored:

Corona has affected them because they were playing as a group outside. . . So now you know he plays alone in the house so he is bored. This social distance thing. . . .he isn't playing anymore–IDI9, mother of a four-year-old

As noted earlier, restrictions placed considerable strain on parents'/carers' mental wellbeing, and this context was described as having an impact on the parents' ability to provide responsive care:

I locked myself in the house and it reached a point and I said I better get sick with Corona instead of seeing how children are crying daily. . . [My 6-month old] would cry for the whole day but what can we do?–IDI5

Most childcare provision was reported to have closed when schools did. This was a result of several factors, including reduced demand because newly unemployed parents or older siblings (whose schools had closed) could now play a larger role in providing childcare at home, or because of parental concern about transmission risks in childcare:

[Paid childcare] was not going on because you are fearing to take your child and meet with other children. Every parent was making the children fearful.–IDI7

There were no jobs. . . we were not going [to paid childcare] when the schools were closed. All of us, even the children, were playing with him–IDI5

Parents/carers also reported cutting back on purchases of books, toys or clothes for the family, and being unable to afford school fees when schools reopened, in some cases leading to children being moved into cheaper schools when lockdown ended:

It even became hard to pay for school. They were going to a good school so I had to transfer them–IDI21, uncle of a three-year old who used to attend childcare

When children were allowed to play, either alone, with family members, or with others in the community, this was also described as insufficient and leading to learning losses or lowered school-readiness amongst those due to be starting school:

You see during that time they were playing a lot. . . just playing. Playing is good but she was not reading at all. So, the things she had learnt, the teacher [at the pre-school] had to teach her again so she can catch up. . . she lagged a little.–IDI8, mother and former user of childcare for now school-aged child

**Security and safety.** When asked about levels and types of violence in their communities during this period, most study participants reported that Covid-19, and the 'lockdowns', led to an increase in the level of crime and domestic violence in their communities, or even in their own households. This worsening of community safety was described as being associated with economic hardship and food insecurity:

[Domestic violence] is not far. . . even in my house. We are struggling a lot because of money. . . Because of money. One thing that makes people violent is money. Lack of money causes people to be violent. . . . . .when you have fifty shillings they see as if you have hidden another fifty shillings in your pocket. Such things, problems, is what makes people violent. Poverty.–IDI5

There have been cases [of domestic violence]. You know, when people lack money mostly they disagree. . . So you can come and you were not successful to bring money for the day. . . you find they disagree and you fight. You know they are non-permanent houses–we hear people. Maybe they have disagreed because of money. Maybe one needs food and they haven't provided. For that food they fight.–IDI17

In interviews, perhaps due to the sensitive nature of this topic and the taboos which sometimes affect it, respondents did not report if or how this increase in domestic violence affected children.

**Help, when available, mostly came from within the community.** One final major theme was that help, when provided, largely came from within the community. One example of this was informal sharing of limited food amongst neighbours:

> You visit the neighbour and eat the food they have. Or sometimes you hustle and go to borrow food like that. It may not be the immediate neighbour here you can go out and go on the other side and meet a friend–IDI7

In addition, flexibility in rent payments, including reductions or deferral of payments was important to many. Flexibility amongst landlords was described as being variable, with those who were known and trusted being seen as more willing to help. There were also examples of community leaders (chiefs) applying pressure to landlords to reduce the risk of evictions:

> Landlord has helped a lot because up until now we have not paid. But there is something that came and helped the people in iron sheet houses. . . Chief came and said people shouldn't be pressured a lot about the houses–IDI5

Some respondents did talk about external assistance, including one who described benefiting from the government cash-for-work programme (Kazi kwa Vijana—Kenya Youth Empowerment Programme), alongside food assistance, but this seemed to be uncommon and infrequent.

Overall, most experiences were relatively universal, with a consistent sense of the pandemic leading to a worsening of living conditions. However, and in contrast to most of the respondents, a small number of participants talked about how day-to-day life had in fact remained quite consistent. This was either because their own work hadn't changed, or because life in the slums was very hard even before the pandemic. In addition, some described how they had managed to 'shield' their child(ren) from impacts of the pandemic:

> Because we are the ones struggling so she can get what she needs . . . her life is going on as usual–IDI2

## Discussion

This research suggests that, despite being a low risk from SARS-CoV-2 infection, the Covid-19 pandemic has radically and negatively affected the care of young children in Nairobi slums, largely due to the direct and indirect effects of pandemic restrictions. The impacts are strikingly broad, affecting all domains of nurturing care, and deep, in terms of the scale of especially economic hardships.

### Key findings in context

Our findings are consistent with much other published research yet we also provide additional child-centred insights and a richer description of how policies–which were often broadly applied–affected people living in slums specifically.

Considering the **cross-cutting** economic impacts, Oyando et al. [19] found, through telephone surveys, that people reported, economic and social disruption across three counties in

Kenya, with especially pronounced effects on income, and amongst the poorest. However, these were only reported generally, without any detail on how these economic disruptions affected peoples' day to day lives, and especially those of families with young children.

On **nutrition**, Kansiime et al. [20] looked specifically at food security impacts of Covid-19 across Kenya and Uganda, and found that more than two thirds (of a cohort completing an online survey) experienced income shocks and worsened food security. Kimani-Murage et al. [21] concluded that restrictive Covid-19 control measures exacerbated the pre-existing vulnerability to food insecurity amongst the urban poor and violated their human right to food; findings consistent with the descriptions of widespread exacerbation of food insecurity amongst respondents in this study.

We found that Covid-19 led to little reported direct harm to respondents, at least that they were aware of, but considerable disruption to **health** services. Oluoch-Aridi et al. [22] conducted a qualitative study looking specifically at the impact of the pandemic on maternity services in Nairobi and identified many themes consistent with ours, including high levels of concern and perceptions of risk early in the pandemic, some reported reductions in access to maternal healthcare alongside significant economic harms including worsening food insecurity due to lockdowns and curfews. Ahmed et al. [23] also explored the impacts of Covid-19 on access to healthcare across seven slums around the world, noting reduced access to services, increases in costs and fear discouraging utilisation; something we heard described by respondents.

A mixed-methods assessment of the health effects of Covid-19 in Kenya found significant reductions in outpatient visits and–in keeping with the worsening community **safety** and increases in **domestic violence** reported in our study–an increase in sexual violence cases reported [24]. The gendered aspects of the impacts extend beyond violence, however, as we found and as has also been noted by others in terms of the socio-economic impacts disproportionally affecting women and girls [25, 26].

A recently published systematic review of the effects of Covid-19 on nurturing care around the world found an evidence base that was limited and biased towards high-income settings, and which suggested that Covid-19 would lead to a need for increased support for young children to thrive in the pandemic [27]. Particular priorities identified were the need to address parent/caregiver stress, burnout or depression and the potential for knock-on harsher parenting affecting **responsive caregiving and early learning**. The authors also identified a risk of reduced child safeguarding referrals and an urgent need for further research, including qualitative studies, to understand these risks in more depth. The absence of references to harsher parenting in our results probably reflects both the fact that we did not directly explore this issue and taboos around this subject.

Many of these impacts identified are consistent with our findings, although it is notable how few of these studies have explicitly considered how the pandemic has affected young children specifically, despite their biological and social vulnerability. For example, although considerable attention has been paid to pandemic-related disruption to education systems in and beyond Kenya [28, 29], much less attention has been paid to the disruption to childcare services and how household tension at times affect the care of young children. This is despite the fact that, as noted earlier, concerns were raised about the risks to young children by some before, during and after implementation of stringent Covid-19 control measures in Kenya [8] and beyond [30, 31]. Despite these warnings, our study suggests that only limited, and largely community drawn, support was received by families, suggesting that only limited policy attention and resources were devoted to these issues.

Overall, our findings suggest that concerns about the risks of 'lockdown' to young children in slums were largely well founded. Multiple harms or negative impacts of stringent Covid-19

control measures on vulnerable young children growing up in slums were reported, and these spanned all domains of nurturing care. It also seems likely that many of these impacts affected families more generally, including those with older children.

## Strengths and limitations

There are a number of strengths to this study. Firstly, in-depth interviews allowed a deep exploration of parental perspectives and experiences during the pandemic. A purposively selected sample meant that a variety of parents/carers were interviewed, including male and female carers and those looking after different ages of children. This allowed gendered differences to be identified, although in general key themes were largely consistent across these groups. We were able to collect high quality data through interviews being conducted by an experienced researcher (RM) combined with regular analytical and reflexivity meetings.

A limitation of the research was, due to the prevailing Covid-19 control measures, the use of an existing sampling frame which was based on prior, albeit recent, in-person enumeration of potential telephone survey respondents by our data collection partner. We were also unable to remotely interview children themselves to ask them directly about their experiences. In addition, we were initially concerned that using remote data collection (telephone interviews) on occasions would present challenges to building rapport, although in practice telephone interviews worked better than we expected, with good rapport being built and few dropped calls.

## Unanswered questions and future research

The impacts reported by participants in this study ought to be explored in more detail, including through efforts to quantify their distribution and magnitude, and resultant impacts on child health and development. The NECS Covid-19 impacts tracker has tracked disruption to early childhood services over time in Nairobi (manuscript in preparation), but studies from a variety of settings are urgently needed, including those that include measurement of child health and development outcomes.

Only through such research, alongside concurrent efforts to assess the real-life benefits of different Covid-19 control measures, can an informed discussion about the overall case for these types of pandemic control measures, in particular stringent 'lockdowns', be considered. Such analyses should inform the response to both future SARS-CoV-2 waves and other emergencies [30].

These results also imply an urgent need for both economic support and broader investment in public health and wellbeing for those living in slums, including in emergencies including epidemic disease outbreaks [32]. Crucially, such investments are likely to be needed both in the short- and long-term to try to mitigate short-term risks like food insecurity, and to ameliorate some of the longer-term harms including to early childhood development and education. In addition, longer term investments in preparation for future crises are also needed [33].

## Conclusion

Based on the experiences of parents/carers, the Covid-19 pandemic, and efforts to control it, appear to have exacerbated adversity amongst young children growing up in slums in Nairobi. This includes through disrupting fragile and weak health, education, childcare and (largely informal) employment systems, and through this placing considerable economic and social distress on vulnerable families and communities. Consideration of these insights can help to inform mitigation efforts and future epidemic control policy discussions. They imply that if blunt policy instruments like 'lockdowns' are to be used at all, then considerable efforts ought

to be made to mitigate their associated harms, especially to young children growing up in informal settlements.

## Supporting information

**S1 Appendix. IDI consent script and topic guide.**
(DOCX)

**S1 Fig. Illustration of inductive and deductive codes.**
(JPG)

**S1 Table. Characteristics of IDI participants.**
(DOCX)

## Acknowledgments

We would like to acknowledge and thank the study participants for the time and insights that they shared with us to conduct this study, and to thank Antonio Aparicio at LSHTM and Pauline Ochieng at APRHC for vital administrative support to the study. We would also like to thank the reviewers for their helpful suggestions which we feel have strengthened the paper considerably.

## Author Contributions

**Conceptualization:** Robert C. Hughes, Sunil S. Bhopal, Betty R. Kirkwood, Zelee Hill, Patricia Kitsao-Wekulo.

**Data curation:** Robert C. Hughes, Ruth Muendo, Silas Onyango, Patricia Kitsao-Wekulo.

**Formal analysis:** Robert C. Hughes, Ruth Muendo, Silas Onyango, Zelee Hill, Patricia Kitsao-Wekulo.

**Funding acquisition:** Robert C. Hughes.

**Investigation:** Robert C. Hughes, Ruth Muendo, Sunil S. Bhopal, Silas Onyango, Zelee Hill, Patricia Kitsao-Wekulo.

**Methodology:** Robert C. Hughes, Sunil S. Bhopal, Zelee Hill, Patricia Kitsao-Wekulo.

**Project administration:** Robert C. Hughes, Silas Onyango.

**Supervision:** Sunil S. Bhopal, Elizabeth Kimani-Murage, Betty R. Kirkwood, Zelee Hill, Patricia Kitsao-Wekulo.

**Writing – original draft:** Robert C. Hughes.

**Writing – review & editing:** Robert C. Hughes, Ruth Muendo, Sunil S. Bhopal, Silas Onyango, Elizabeth Kimani-Murage, Betty R. Kirkwood, Zelee Hill, Patricia Kitsao-Wekulo.

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
