## [Decision Letter · Decision Letter 0]

3 Jan 2023

PGPH-D-22-01455

Parental experiences of the impacts of COVID-19 on the care of young children; qualitative interview findings from the Nairobi Early Childcare in Slums (NECS) Project

Dear Dr. Hughes,

Thank you for submitting your manuscript to PLOS Global Public Health. After careful consideration, we feel that it has merit but does not fully meet PLOS Global Public Health’s publication criteria as it currently stands. Therefore, we invite you to submit a revised version of the manuscript that addresses the points raised during the review process.

Editor Comments: 

The reviewers and I appreciate the potential for impact that this manuscript has, by conducting qualitative interviews to elucidate parental experiences during COVID-19 in low-income communities in Kenya. The manuscript is generally well-written and raises important questions. It also builds on and extends previous work in the communities in important ways.As noted by the reviewers, the methods need additional explanation. Given the prominence of the Nurturing Care Framework to the study design and analysis, it would be helpful to describe this framework in the introduction (perhaps a figure might also be useful), to help the readers understand how the framework informed the interview guide development and coding. Additional details on the interview guide development and revisions should be added, especially in terms of whether the guide was piloted, translated and back-translated, and how and why the guide was revised. I would also encourage the authors to include their final interview guide and codebook as appendices/supplemental materials. The comment that a phenomenological approach was used is a little confusing. I did not see reference to bracketing in the analysis, nor was it clear how deep, lived experience might be understood by a 15 minute phone interview--perhaps the authors also included data from the parent study to gain a deeper understanding of participants' experiences? If so, this should be made clearer. Further, the research is strongly driven by an external framework (the Nurturing Child Framework), which also seems somewhat counter to a phenomenological approach. I think the pragmatic approach makes more sense, unless there is additional information that I missed.I also agree with Reviewer 2 that some of the assertions made in the discussion (child-centered data, gendered differences) are not very clear in the results. In addition, the links between the Nurturing Care Framework and the themes could also use additional explanation.  I would also encourage the authors to consider Reviewer 1's questions related to the potential benefits/utility of the research to the participants, and to be a little clearer about the community engagement piece. If the community engagement methods and outcomes are outside the scope of the manuscript, then perhaps these are already described in a separate paper or could be added to the supplementary materials. I also wondered whether research participants received any financial incentive or phone airtime for their participation.

We look forward to receiving your revised manuscript.

Kind regards,

Marie A. Brault, PhD

Academic Editor

Journal Requirements:

a. State what role the funders took in the study. If the funders had no role in your study, please state: “The funders had no role in study design, data collection and analysis, decision to publish, or preparation of the manuscript.”

b. If any authors received a salary from any of your funders, please state which authors and which funders.

2. We have noticed that you have uploaded Supporting Information files, but you have not included a list of legends. Please add a full list of legends for your Supporting Information files after the references list. 

3. In the online submission form, you indicated that "We are happy to make available to all qualified researchers on request the underlying data drawn on for this research. Given the challenges of genuine anonymisation of qualitative data, we have not posted the data to an open repository, but we are happy to facilitate sharing requests via the corresponding author.". All PLOS journals now require all data underlying the findings described in their manuscript to be freely available to other researchers, either 1. In a public repository, 2. Within the manuscript itself, or 3. Uploaded as supplementary information.

Additional Editor Comments (if provided): See above.

Reviewers' comments:

Reviewer's Responses to Questions

**Comments to the Author**

1. Does this manuscript meet PLOS Global Public Health’s publication criteria? Is the manuscript technically sound, and do the data support the conclusions? The manuscript must describe methodologically and ethically rigorous research with conclusions that are appropriately drawn based on the data presented.

Reviewer #1: Yes

Reviewer #2: Partly

2. Has the statistical analysis been performed appropriately and rigorously?

Reviewer #1: Yes

Reviewer #2: N/A

3. Have the authors made all data underlying the findings in their manuscript fully available (please refer to the Data Availability Statement at the start of the manuscript PDF file)?

Reviewer #1: Yes

Reviewer #2: Yes

4. Is the manuscript presented in an intelligible fashion and written in standard English?

Reviewer #1: Yes

Reviewer #2: Yes

5. Review Comments to the Author

Reviewer #1: This article is an offshoot of a wider project, begun before covid, about childcare in Nairobi townships. The authors used their database and contacts to carry out additional research about the impact of covid on households with young children. Whilst the research methods are impeccable, and the exposition is very clearly stated, this article raises the ethical questions put forward by, for example, Seye Abimbola about foreign gaze and structural inequities. The research reveals the shocking circumstances of families barely surviving in their day to day life. But who is such research knowledge for, and where is it being reproduced? What is the benefit to the participants of having participated? The article mentions "community involvement" and "community meetings" but gives not details. Instead, the authors state that "Consideration of these insights can help to inform mitigation efforts and future epidemic control policy discussions." If this is the main purpose of the research, how is this to be done? Who with? What role will those interviewed play in future mitigation efforts, other than having given information about their circumstances? I would be happier with this article if the ethics of a mainly white Western Research Institution (LSHTM) working with extremely deprived women and the cliches about policy discussions, were more stringently explored.

Reviewer #2: Introduction

• Please make sure you use the same word consistently e.g. covid, Covid-19, covid-19, covid19, etc

• This needs more detail on the specific public health and social measures that were enforced as they differed in each country. Was the focus on social distancing, re-direction of public healthcare services to COVID-19, closure of business, restriction of movements, etc. The kind of measures that were enforced will be associated with the impact and can therefore frame these impacts.

• What were the impacts of covid measures on social determinants of health in general in Nairobi/Kenya?

• Specify the age for early childhood

• What were the direct impacts of Covid-19 on children in Nairobi/Kenya/Africa? Please give a few lines on findings from research on covid impact on children in Nairobi/Kenya/Africa.

• The word “slum” can be considered derogatory and stigmatising. Please replace it with a non-stigmatising word.

Methods

• The parent study needs to be introduced here and the link between both studies.

• Specify which ethnicities

• what was the sample size and how did you decide on this sample size e.g. Malterud's information power

• line 149: reference for the literature review. Is this an existing literature review or was the review conducted by the authors of this article prior to the study?

• Introduce the framework in the introduction

• Line 158: is this the NECS study? Needs more detail.

• What was included in the information script?

• Did all participants consent? Could participants decline to participate? How many declined?

• Please include a section on researcher reflexivity i.e. critical reflection about the position you are taking as a researcher, and the influence of your specific background, education, race, gender, socio-economic status, etc in relation to the target group in your research, and how you have taken this stance into account in your research.

• Also include CREDIT author statement.

Results

• Line 234: reported direct effects

• Please give some detail on the reported direct effects as an introduction. Which direct effects did the study ask about and what was the response?

• Line 272: Was this not already the situation before COVID-19? Were men now more responsible for earning money? This doesn't really show that the effects were gendered. Life before Covid-19 was most likely already divided along gender lines. How did it change during Covid-19? Did these divisions become more entrenched? And how exactly?

• Separate the “early learning” and give it its own section.

• Did the parents/carers report how this increase in domestic violence affected the children? For example, did children show an increase in behavioural issues or ran away from home or started bed wetting, etc? How did it affect the children?

• What are your additional insights? Point them out specifically. Right now, this section reads as if you only found what others have also found. What did your study add?

• Another limitation is that children themselves were not interviewed. How did they experience the restrictions?

• What do you mean by fragile and weak systems? Are you referring to the healthcare system, the informal economy, social services, the schooling system, etc? Specify what you are referring to.

6. PLOS authors have the option to publish the peer review history of their article (what does this mean?). If published, this will include your full peer review and any attached files.

**Do you want your identity to be public for this peer review?** For information about this choice, including consent withdrawal, please see our Privacy Policy.

Reviewer #1: No

Reviewer #2: **Yes: **Lieve Vanleeuw

---

## [Decision Letter · Decision Letter 1]

4 Apr 2023

PGPH-D-22-01455R1

Parental experiences of the impacts of Covid-19 on the care of young children; qualitative interview findings from the Nairobi Early Childcare in Slums (NECS) Project

Dear Dr. Hughes,

Thank you for submitting your manuscript to PLOS Global Public Health. After careful consideration, we feel that it has merit but does not fully meet PLOS Global Public Health’s publication criteria as it currently stands. Therefore, we invite you to submit a revised version of the manuscript that addresses the points raised during the review process.

Editor Comments:

I appreciate the authors' engagement with the reviewers' comments. The manuscript is much improved and will make an important contribution. There are just two minor edits that I would encourage the authors to address before acceptance. First, instead of using participant ID numbers to differentiate statements made by different participants, I think it would be more helpful to provide basic demographic information (for example, male caregiver, 30 years old, etc.).Second, Reviewer 3 raises a question about the age of the children and behaviors discussed by parents. This is not a major revision, but might be helpful to clarify if possible.

We look forward to receiving your revised manuscript.

Kind regards,

Marie A. Brault, PhD

Academic Editor

Journal Requirements:

Additional Editor Comments (if provided):

Reviewers' comments:

Reviewer's Responses to Questions

**Comments to the Author**

1. If the authors have adequately addressed your comments raised in a previous round of review and you feel that this manuscript is now acceptable for publication, you may indicate that here to bypass the “Comments to the Author” section, enter your conflict of interest statement in the “Confidential to Editor” section, and submit your "Accept" recommendation.

Reviewer #2: All comments have been addressed

Reviewer #3: (No Response)

2. Does this manuscript meet PLOS Global Public Health’s publication criteria? Is the manuscript technically sound, and do the data support the conclusions? The manuscript must describe methodologically and ethically rigorous research with conclusions that are appropriately drawn based on the data presented.

Reviewer #2: Yes

Reviewer #3: Yes

3. Has the statistical analysis been performed appropriately and rigorously?

Reviewer #2: N/A

Reviewer #3: N/A

4. Have the authors made all data underlying the findings in their manuscript fully available (please refer to the Data Availability Statement at the start of the manuscript PDF file)?

Reviewer #2: Yes

Reviewer #3: No

5. Is the manuscript presented in an intelligible fashion and written in standard English?

Reviewer #2: Yes

Reviewer #3: Yes

6. Review Comments to the Author

Reviewer #2: no further comments

Reviewer #3: Although with relatively little data (21 participants who were interviewed for between 14 and 39 minutes) this is a well conducted and reported study. Please note- I didn't review the earlier version.

I was surprised that the authors were focused on young children under 5, as many of the quotes seemed to be about life in general, or about school aged children, for example "Playing is good but she was not reading at all. So, the things she had learnt, the teacher had to teach her again so she can catch up… she lagged a little." There are not many children who do a lot of reading before their 5th birthday, so strange to choose this quote, rather than anything about delayed language learning, development of confidence, experience of the world etc.

Similarly:

"Corona has affected them because they were playing as a group outside… So now you know he plays alone in the house so he is bored." I wouldn't have thought many little children (aged 0-2) would be playing in a group outside, pre-covid, or that they would be particularly bored inside.

There was nothing that seemed related to having babies or young toddlers at home (for example, if there was no work outside the house, did this give opportunities for longer breast-feeding?), but given this was within a larger study on paid childcare, perhaps the focus was actually on pre-school children i.e.: aged 3-5? Rather than <5? Or is it possible that parents were recruited based on their younger child, but ended up discussing more about the ways the lockdowns were impacting their older school-aged children (perhaps because the impacts were bigger on those older children).

This is the only concern I had. Otherwise this looks good for publication.

7. PLOS authors have the option to publish the peer review history of their article (what does this mean?). If published, this will include your full peer review and any attached files.

**Do you want your identity to be public for this peer review?** For information about this choice, including consent withdrawal, please see our Privacy Policy.

Reviewer #2: **Yes: **Lieve Vanleeuw

Reviewer #3: No

---

## [Editor Report · Decision Letter 2]

17 Apr 2023

Parental experiences of the impacts of Covid-19 on the care of young children; qualitative interview findings from the Nairobi Early Childcare in Slums (NECS) Project

PGPH-D-22-01455R2

Dear Dr. Hughes,

We are pleased to inform you that your manuscript 'Parental experiences of the impacts of Covid-19 on the care of young children; qualitative interview findings from the Nairobi Early Childcare in Slums (NECS) Project' has been provisionally accepted for publication in PLOS Global Public Health.

Best regards,

Marie A. Brault, PhD

Academic Editor